# Numerical Models for Monitoring and Forecasting Ocean Biogeochemistry: a short description of present status

Gianpiero Cossarini[1], Andy Moore[2], Stefano Ciavatta[3], Katja Fennel[4]

[1]Istituto Nazionale di Oceanografia e di Geofisica Sperimentale, Department of Oceanography, Trieste, Italy
[2]Physical & Biological Sciences Division, Ocean Sciences Department Institute of Marine Sciences, Institute of Marine Sciences, University of California Santa Cruz, Santa Cruz, California, USA
[3]Mercator Ocean International, Toulouse, France
[4]Department of Oceanography, Dalhousie University, Halifax, NS, Canada

*Correspondence to*: Gianpiero Cossarini, gcossarini@ogs.it

**Abstract.** The ability to model biogeochemical features in the ocean is a key factor in predicting the health of the ocean: it involves the representation of processes and cycles of chemical elements (such as carbon, nutrients and oxygen) and the dynamics of living organisms such as phytoplankton, zooplankton and bacteria. This paper gives an overview of the main
modelling aspects aimed at describing the low trophic levels of marine ecosystems and shows how they can be coupled with advection and diffusion models. The complexity of biogeochemical models can vary considerably depending on the topics of interest, assumed hypotheses and simplifications of the numerical parameterizations. The paper also discusses the uncertainties in the numerical solution due to the lack of knowledge about the parameterizations, the initial and boundary conditions, the lack of a robust observation network and the high computational cost of running such models.

## 1 Introduction

Marine biogeochemistry refers to the cycling of chemical elements (e.g., carbon, nutrients, oxygen) resulting from physical transport, chemical reactions, uptake and processing by living organisms (e.g., phytoplankton, zooplankton, bacteria). Biogeochemical models describe the low trophic levels of marine ecosystems and are usually coupled with advection and diffusion models. Operational biogeochemical models have generally been developed by incorporating biogeochemical models
developed for research and process-based studies into existing physical forecasting systems (Gehlen et al., 2015) and are used to assess (i) past and current marine ecosystem states and trends and (ii) short-term (days to weeks) or seasonal (months) forecasts (Le Traon et al., 2019; Fennel et al., 2019). When observations are assimilated, simulations of a past period are called reanalysis, while unconstrained simulations of a past period are called hindcast. When the simulations are carried forward to the present, they are referred to as nowcasts. (Fennel et al., 2023). By providing timely information for the current state and a
consistent reconstruction of the past, biogeochemical operational models can support ocean carbon sequestration and storage estimations, monitoring effects of acidification and deoxygenation, marine spatial planning, and, as input for habitat and food web modelling, marine biodiversity conservation and fisheries management.

Biogeochemical models can have a wide range of complexity, from a single nutrient and simple parameterizations of processes to fully explicit representations of several nutrients, trophic levels, and functional groups. They can also include carbonate systems, pollutants (e.g. Hg, persistent organic pollutants -POPs) and other features, depending on the specific goals and domains for which they were developed. This chapter provides a brief introduction to biogeochemical modelling in the context of operational oceanography, more detailed descriptions and discussions can be found in the following articles (Gutknecht et al., 2022; Fennel et al., 2022; Ford et al., 2018). The focus is on the levels of model complexity in existing biogeochemical prediction systems, with examples taken mainly from the Copernicus Marine Service (Le Traon et al., 2019).

## 2.1 Formulations, processes and elements of biogeochemical models

In general, biogeochemical models solve a system of partial differential equations Eq. (1) describing the rate of change of a set of state variables C representing biogeochemical tracers: dissolved inorganic substances (e.g., nutrients), living organic compartments (e.g., primary producers and secondary consumers), and nonliving organic dissolved and particulate matter (Gutknecht et al., 2022; Fennel et al., 2022). The first three terms on the right side of Eq. (1) represent the physical terms: advection (first term), diffusion (second - horizontal - and third - vertical - terms) of biogeochemical tracers, where $K_H$ and $K_V$ are the horizontal and vertical diffusivities, respectively, which act on different spatial scales. The remaining terms describe the sinking processes that affect biological particles (fourth term) and biogeochemical reactions (fifth term).

$$\frac{\partial C}{\partial t} = -v \cdot \nabla(C) + \nabla_H\big(K_H \nabla_H(C)\big) + \frac{\partial}{\partial z}\Big(K_V \frac{\partial C}{\partial z}\Big) + w_{sink}\frac{\partial C}{\partial z} + R_{bio}(T, light, \rho, C) \quad (Eq.\,1)$$

The last term, $R_{bio}$, represents the local source-minus-sink terms for the biogeochemical tracers and is typically based on the principle of conservation of mass to simulate the cycling of chemical elements through various marine compartments. Biogeochemical models (Eq.1) are generally discretized on a grid covering a spatial region of interest and they are solved numerically by using appropriate initial and boundary conditions for each of the tracers. The physical parts of Eq.(1) can be solved directly by the advective-diffusion component of ocean dynamic models (i.e., on-line coupling). Alternatively, the output of the ocean dynamics model is used to force the biogeochemistry off-line (Heinze and Gehlen, 2013). Different schemes can be used to couple the physical and biogeochemical processes to optimize accuracy and computational cost (Bruggeman and Bolding, 2014; Cossarini et al., 2017). Operational biogeochemical models include also data assimilation schemes (Brasseur et al., 2009; Fennel et al., 2019)., with satellite observations being the most commonly used due to their spatial coverage and time availability, even in near real time. Ocean colour chlorophyll is the variable most typically assimilated in biogeochemical models (Nerger and Gregg, 2008; Ciavatta et al., 2011; Fontana et al., 2013; Teruzzi et al., 2014; Ciavatta et al., 2016) but other remote sensing variables have also been tested: diffuse attenuation coefficient (Ciavatta et al., 2014), phytoplankton functional type chlorophyll (Ciavatta et al., 2018; Skákala et al., 2018, Pradhan et al. 2020) and inherent optical properties (Jones et al., 2016). While ocean colour provides unique information about the surface of the ocean, the transfer of surface information to deeper layers usually requires approximations. The emergence of autonomous underwater

sensors (biogeochemical gliders and Argo floats) has opened the possibility to better constrain biogeochemical dynamics below the water surface (Verdy and Mazloff, 2017; Cossarini et al., 2019; Teruzzi et al., 2021; Skákala et al., 2021). Oxygen, chlorophyll and nitrate profiles are currently used in forecast systems for assimilation (Amadio et al., 2024), but also for parameters tuning (Wang et al., 2021, Yumruktepe et al., 2023, Falls et al., 2022), validation of operational systems (Salon et al., 2019; Mignot et al., 2023) and adaptive monitoring of phytoplankton blooms (Ford et al., 2022).

Unlike physical models based on Navier-Stokes equations (Bell et al., 2024), there are no fundamental laws and principles for the biogeochemical term ($R_{bio}$). Rather, equations describing biogeochemical processes rely on empirical relationships based on laboratory experiments (e.g., nutrient limitation experiments, grazing dilution experiments), biological theories, and ecological principles based also on biogeographic relationships.

The NPZD approach, which stands for nutrient-phytoplankton-zooplankton-detritus (Fasham et al., 1990), is the basis of most

marine biogeochemical models. In its simplest form, the cycling of a single nutrient (e.g., nitrogen) is represented by four marine compartments: inorganic nutrient and living (phytoplankton and zooplankton) and nonliving (detritus) organic compartments. The nutrient fluxes between the compartments are the uptake of the nutrient as a function of phytoplankton growth, the mortality and grazing of phytoplankton and zooplankton, and the remineralization of the detritus compartment. In the original Fasham model (Fasham et al., 1990), the nutrient inorganic pool (nitrogen) is divided into ammonium and nitrate,

and the remineralization process includes bacteria and dissolved organic nitrogen, increasing the number of state variables from four to seven and nearly doubling the number of processes described.

A schematic representation of the cycles of multiple chemical elements (e.g., nutrients) among living and non-living compartments together with some additional features presented below is shown in Figure 1.

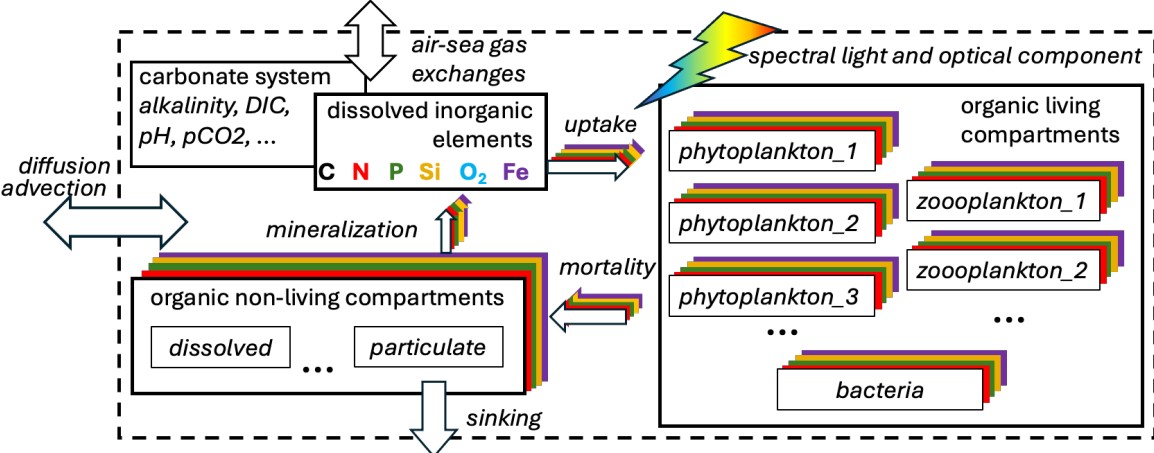

Figure 1: simplified scheme of compartments (boxes) and processes (arrows) representing the cycles of multiple chemical elements typically included in a biogeochemical model. The symbol ... indicates the increase of the number of compartments and variables (e.g., multiple plankton functional types, multiple size compartment of non-living organic matter or additional variables resolved by the carbonate system.

Increasing model complexity (e.g., greater number of state variables and processes) enables the expansion of the model objectives and the range of applications, but is accompanied by larger uncertainties in the parameterization and higher computational costs. There is no general consensus on the level of complexity of biogeochemical models and the priority for new components to be added. This often depends on the specific objectives for which a model is being built. In recent years, the complexity of biogeochemical models used in operational oceanography has increased, as have their applications. These span multiple objectives: monitoring ocean state and variability, assessing ocean health (acidification, eutrophication, deoxygenation), supporting resource management, and studying pollutant impacts.

The number of chemical elements is often increased, to include typically carbon, other macronutrients such as phosphorus and silicon, and micronutrients (e.g., iron). The increase in model complexity allows modellers to represent a wider range of chemical and biological processes such as nitrification, denitrification, calcification, competition for the limiting nutrients and dimethylsulfide (DMS) dynamics. Another typical tracer included in biogeochemical models is oxygen because of its importance for ocean health (e.g., deoxygenation, Schmidtko et al., 2017; Gregoire et al., 2021; Bopp et al., 2013) and the effects of low oxygen concentrations (hypoxia) in changing ecosystem functions (Baird et al., 2004).

Fixed or variable nutrient stoichiometry can then be formulated within the simulated organisms, e.g. phytoplankton. Typical values of fixed nutrient ratios are $138[O2] : 106[C] : 16[N] : 15[Si] : 1[P] : 0.1-0.001[Fe]$ (Redfield, 1934; Lenton and Watson, 2000). When models include variable stoichiometry (e.g., Vichi et al., 2017; Tagliabue et al., 2011), multiple state variables are required to represent the living organic compartments, and a formulation of intracellular ratios can be used to simulate the multiple nutrient limitation of phytoplankton growth (Klausmeier et al., 2004). Primary production, the basis of the marine food web, is the chemical synthesis of organic compounds from dissolved carbon dioxide through chlorophyll-mediated photosynthesis. When chlorophyll is explicitly included in models, photosynthesis and acclimation to light can be dynamically simulated to balance growth rate and variable chlorophyll-carbon ratio as a function of light, nutrient limitation, and temperature (Geider et al., 1997).

Complexity of biogeochemical models can be measured by the number of plankton functional types (PFTs) used to simulate the trophic food web. The autotrophic community can be conceptually grouped considering various ecological functions (e.g., silicifiers, calcifiers, nitrogen fixers, and dimethylsulfide -DMS- producers), cell size (e.g., pico-, nano-, and micro-phytoplankton), and specific physiological traits such as optical absorption, light use, growth rate and affinity for nutrients (Hood et al., 2006). To improve the representation of the dynamics of phytoplankton functional groups, biogeochemical models can include a spectral radiative component which resolves solar radiation penetration in the water column (Dutkiewicz et al., 2009; Skákala et al., 2020; Álvarez et al., 2022). The zooplankton community can be subdivided by size (nano-, micro-, meso-, or macro-) and grazing strategy (herbivorous versus carnivorous. Additionally, a rigid partition between autotrophs and heterotrophs is not exhaustive, and the food web can incorporate mixotrophs to account for organisms that obtain energy through both photosynthesis and consumption of others (Flynn et al., 2013; Mitra et al., 2014).

Biogeochemical models can target biodiversity (Litchman and Klausmeier, 2008) if the number of functional plankton groups is large enough to deterministically represent niches based on certain factors (e.g., adaptation to the light spectrum; Álvarez et al., 2022), or by including tens or hundreds of PFTs with randomly prescribed parameters so that the fittest groups can prevail in the resulting ecosystem (Follows et al., 2007).

Assessing ocean carbon uptake and the associated ocean acidification requires modeling of the marine carbonate system: the two prognostic variables are typically dissolved inorganic carbon and alkalinity, and carbonate chemistry is solved to determine water acidity and to calculate the air-sea $CO_2$ gas exchange (Zeebe and Wolf-Gladrow, 2001; Artioli et al., 2012; Cossarini et al., 2015a).

The microbial loop describes the role of bacteria in decomposing organic matter that is converted back to nutrients. It also includes the channelling of energy and matter to higher trophic levels by microzooplankton, which can be an important pathway in oligotrophic conditions (Legendre and Rassoulzadegan, 1995; Hood et al., 2006). In addition, models can describe the dynamics of multiple pools of dissolved organic matter (e.g., labile, semilabile, and refractory) characterized by turnover timescales ranging from days to years (Anderson et al., 2015; Glibert and Mitra, 2022). In coastal and shallow water applications, a benthic model allows to represent the mutual interaction and nutrients, carbon and oxygen exchanges between the water column and the sediment (Soetaert et al., 2000).

Biogeochemical models can be linked or coupled to higher trophic level or ecosystem models (Libralato, 2024). This requires parameterization or explicit representation of the link of phytoplankton productivity and zooplankton mortality with higher trophic level (HTL) compartments such as nekton and fishes, and possibly the feedbacks from HTL to biogeochemical processes (e.g., Travers et al., 2009).

Although not yet ready to be integrated into an operational prediction system, additional - and useful to society - components of biogeochemical models could include the fate, dynamics and transport of metals and POPs (Melaku Canu et al., 2015; Wagner et al., 2019, Bieser et al., 2023), including bioaccumulation in low trophic level compartments and cumulative impacts on marine species and ecosystems (Rosati et al., 2022; Zhang et al., 2020).

## 2.2 main models used in operational systems

Unlike ocean dynamics, where a limited number of numerical models are used in operational applications (e.g., NEMO, ROMS, MITGCM, see Alvarez-Fanjul et al., 2022), there is a long list of biogeochemical models that have varying level of complexity in response to specific regions and topics of interest for which were developed. As part of the UN Decade of Ocean Science for Sustainable Development program, the Ocean Prediction Decade Collaborative Center (Alvarez-Fanjul et al., 2024) is promoting the Atlas of Operational Systems, which also describes their biogeochemical component (oceanprediction.org). Some of the biogeochemical models used in operational prediction systems are briefly presented below, roughly ordered by increasing complexity.

HadOCC (Palmer and Totterdell, 2001): a model of low complexity (10 variables) with a single phytoplankton and single zooplankton, fixed stoichiometry used to produce global reanalysis of the carbon cycle (Ford and Barciela, 2017).

SCOBI: used for reanalysis of nutrient cycling in the Baltic Sea (Liu et al, 2017), it has fixed nutrients stoichiometry in three phytoplankton and one zooplankton and includes anaerobic processes and a sediment module for oxygen and nutrient dynamics (Eilola et al., 2009).

NEMURO: (Kishi et al., 2007) a relatively simple low trophic level model of the Pacific Ocean (11 state variables) based on N dynamics with two phytoplankton and two zooplankton, that has been coupled with HLT model (e.g., bioenergetic fish model; Kishi et al., 2011).

ECB: (Feng et al., 2015) developed to study eutrophication in the Chesapeake Bay, it consists of 11 variables (C and N cycles) with one single phytoplankton and single zooplankton and processes applicable for estuarine ecosystems, such as inorganic suspended solid dynamics and the impact on light attenuation (Feng et al., 2015; Irby et al., 2019; Irby et al., 2018).

GulfMexico: developed to investigate eutrophication and acidification in the Gulf of Mexico, it is a model of intermediate complexity (15 variables) that simulates N, P, O2 and C dynamics with a single phytoplankton and single zooplankton group, a sediment-water fluxes parameterization, and the carbonate system (Fennel et al., 2011; Laurent et al., 2017).

PISCES: is a model of intermediate complexity (24 state variables) with five nutrients, fixed-stoichiometry, two phytoplankton and two zooplankton size classes and includes carbonate system and dissolved oxygen dynamics (Aumont et al., 2015). It is currently used in regional (north-eastern Atlantic, Gutknecht et al., 2019) and global operational systems (Mignot et al., 2023). A version with variable stoichiometry (PISCES-QUOTA) also exists and is used for climate scenario studies (Kwiatkowski et al., 2018).

ECOSMO: (Daewel and Schrum, 2013), its operational version in use for Northern Atlantic and Artic Ocean (Yumruktepe et al., 2022) has two phytoplankton, two zooplankton, multiple nutrients (N, P an Si) and fixed molar Redfield ratio but a variable chlorophyll-to-carbon dynamics and it includes a nutrient sediment layer.

ERGOM: used in the Baltic Sea operational system, it is a model of intermediate complexity (25 variables) with three phytoplankton and two zooplankton groups, and includes processes related to hypoxia and anoxia, carbonate system and a radiative model with dynamics for coloured dissolved organic matter (Neumann, 2000; Neumann et al., 2015).

BAMHBI: developed for the Black Sea which is characterized by an anoxic deep layer, the model includes 33 pelagic state variables, with multinutrients, eight plankton functional types and explicitly describes processes in the anoxic layer. It also includes dynamics of the sedimentary stocks of organic C, N, P and biogenic Si (Grégoire et al., 2008; Grégoire and Soetaert, 2010; Capet et al., 2016, Ciliberti et al., 2022)

eReefs/vB3p0: designed for water quality in the Australian Great Barrier marine ecosystem, it is a complex model resolving N, P, C, O2 cycles in pelagic (four phytoplankton and two zooplankton) and sediment (seagrass and coral) environments. It includes carbonate chemistry (Mongin et al., 2016), bio-optics and bleaching (Baird et al., 2016; Baird et al., 2020).

BFM: a multi-nutrient and multi-plankton model (Álvarez et al., 2022) with more than 50 variables, it includes carbonate chemistry (Cossarini et al., 2015b; Cossarini et al., 2017), bio-optics (Lazzari et al., 2021) and pollutants (Rosati et al., 2022) and is currently used in the operational system and reanalysis of the Mediterranean Sea (Salon et al., 2019; Cossarini et al., 2021).

ERSEM: developed for regional (north-eastern Atlantic and North Sea) and global studies, it is a complex model (more than 50 state variables) including multi nutrients, multi-plankton, carbonate system and a sediment layer (Baretta et al., 1995; Butenschön et al., 2016).

DARWIN: a complex multi-nutrient model in which the plankton community comprises hundreds of groups by taking into account cell size and functional traits to study the biodiversity and biogeography (Dutkiewicz et al., 2009).

**3 Conclusions**

Given the complexity of marine ecosystems, the development of biogeochemical models is the result of compromises and simplifications, and no single approach can realistically encompass all relevant aspects of marine ecosystem dynamics. Determining the appropriate level of complexity depends on the specific objectives and supporting information for each application, while standard assessment frameworks (Hernandez et al., 2018) represent essential tools to assess model
performance. Increasing model complexity does not necessarily mean better performance (Xiao and Friedrichs, 2014; Kwiatkowski et al., 2014; Gehlen et al., 2015; Séférian et al., 2020). Indeed, despite recent significant technological advances in observing systems, the lack of biogeochemical observations, both in terms of number of variables and spatiotemporal availability, remains the major obstacle for thorough validation and optimization (e.g., tuning parameters).

In addition to inherent uncertainties in model structures and parameterizations, important sources of uncertainty arise from
numerical solution settings in spatially discretized domains: e.g, initial conditions, lateral conditions for open boundaries, inputs of chemical compounds and suspended matter from rivers and atmospheric deposition, and ocean dynamics driving the transport of biogeochemical tracers. Despite major advances in high performance computing, the computational cost of a model still constitutes an issue when resolution of the numerical solution and complexity (e.g., the number of biogeochemical tracers) are increased. Many of the biogeochemical models have emerged as community models that should guarantee a
distributed and affordable effort to keep them up to date with the advancement in marine ecology knowledge and the requirements of evolving computer science and of the coupling with physical and Earth system models and data assimilation frameworks. Rapidly evolving applications of artificial intelligence in marine biogeochemistry can assist in optimising model parameters, developing hybrid models to improve predictions and operational system efficiency, and detecting patterns in large data sets from reanalysis. Linking microbial community dynamics to ecosystem processes through metagenomic data can
improve models describing nutrient cycling, carbon fluxes and diversity. New coupling paradigms are needed to promote the integration of biogeochemical models with the dynamics of pollutants, high trophic levels and Earth system components.

In addition to science-driven developments, operational biogeochemical systems can evolve to respond to societal demands to assess the impacts of heat waves, oxygen depletion and acidification on marine resources and the role of the oceans in achieving the goal of carbon neutrality.

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

**Competing interests**

The contact author has declared that none of the authors has any competing interests.

**Data and/or code availability**

Not applicable.

**Authors contribution**

CG prepared the first draft, all co-authors participated to review and finalization.
