# Peer review of "Numerical Models for Monitoring and Forecasting Ocean Biogeochemistry: a short description of present status"

_State of the Planet, 2024_

## Referee Comment (RC1)

**Comment on "Numerical Models for Monitoring and Forecasting Ocean Biogeochemistry: a short description of present status"**

Anonymous Referee

**General Comments**

The manuscript provides a comprehensive overview of the current state of biogeochemical modeling in the context of operational oceanography. The authors have successfully outlined the complexity of these models, the various approaches used to couple biogeochemical processes with physical oceanographic models, and the challenges associated with parameterization and computational costs. The manuscript is well-structured, and the use of specific examples from different modeling systems enhances its clarity and relevance. The paper is a valuable contribution to the field, offering insights into the advancements and ongoing challenges in marine biogeochemical modeling.

**Specific Comments**

**1. Introduction:**

The introduction effectively sets the context for the discussion, presenting the main aspects of modelling. However, it would benefit from highlighting the importance of biogeochemical modelling for understanding marine ecosystems, giving more details on the practical applications and implications of these models for marine resource management and policymaking. The reader would like to know why modelling ocean biogeochemistry is important. What is at stake?

**2. Basic Formulation and Processes of Biogeochemical Models:**

The section provides a clear and concise explanation of the fundamental principles behind biogeochemical models. Including a schematic diagram of a typical biogeochemical model could further enhance the readers' understanding.

The discussion on the NPZD approach is thorough, but it would be useful to compare this approach with other modeling frameworks, highlighting their respective strengths and limitations.

The manuscript explains the trade-offs between model complexity and computational cost. It would be beneficial to include a table summarizing the key features and applications of the various models discussed (e.g., HadOCC, SCOBI, NEMURO, etc.).

The section on the incorporation of additional components (e.g., metals, POPs) into biogeochemical models is particularly interesting. More examples of how these components have been successfully integrated and validated in existing models would strengthen this section.

Even if it's not the focus of the paper, the discussion on data assimilation could be developed further. The manuscript could delve deeper into the specific challenges and potential solutions related to assimilating different types of observational data (e.g., satellite data vs. in situ measurements).

The role of emerging technologies, such as autonomous underwater vehicles and biogeochemical Argo floats, is briefly mentioned. Expanding this discussion to include recent advancements and case studies would provide a more comprehensive view.

The title of this section is a little reductive in relation to all the aspects covered. I suggest rewording the title or dividing it into several subsections (ex: basic formulation, coupling, model complexity, data assimilation, main models used...).

**3. Conclusions:**

The conclusions summarize the key points effectively but could be expanded to include a forward-looking perspective on the future of biogeochemical modeling. What are the emerging trends and technologies that could shape this field in the next decade?

Discussing the potential for interdisciplinary collaboration and the integration of biogeochemical models with socio-economic models could provide a broader context for the readers.

**Technical Corrections**

**Introduction:**

- The definition of 'reanalysis', 'hindcast' and 'nowcast' may vary depending on the application, methodology, community or context. A definition of these terms should be provided.

- Line 30, cite some pollutants in brackets to help the reader.

**Equation Formatting:** Ensure that all equation terms are consistently formatted and clearly presented. For example, line 40, KH should be written $K_H$ (with H as a subscript). Please check all terms (KV, Rbio...).

**References**: Please check that all works cited are included in the reference list and that all references are correctly cited in the text. Please also check the spelling of the names mentioned and the year of publication. There are many typos...  This list below is not exhaustive, please check each reference...
-The text refers to Daewl instead of Daewel, Gutkenecht instead of Gutknecht.
-Vichi et al (2007), Vichi (2015), Yumruktepe et al (2023) are cited in the text but not included in the reference list.
-Vichi et al (2017), Yumruktepe et al (2022) are mentioned in the Reference list but are not cited in the text.

**Acronyms**: All acronyms and chemical products should be defined when first used. For example: DMS HTL, POPs, CDOM...

**Typographical Errors**

Line 26: " process-base" should be " process-based".

Line 38: "compartmens" should be "compartments"

line 90: "plankton functional groups" should be "plankton functional types" to be consistent with the acronym PFTs

Line 141, remove the dot before the reference Aumont et al (2015)

line 150, please check: "multinutrients and multi-plankton several", something is wrong

---

## Author Comment (AC1)

**Title: Numerical Models for Monitoring and Forecasting Ocean Biogeochemistry: a short description of present status**
Author(s): Gianpiero Cossarini, Andy Moore, Stefano Ciavatta, Katja Fennel
MS No.: sp-2024-8
Report: Ocean prediction: present status and state of the art

Dear Editor,
We thank the 3 reviewers for their positive comments. Their requests have been carefully considered. In the following reply letter, reviewers' comments are in **bold green**, the answers in black and the proposed revised text in blue.

**RC1**
**General Comments**
**The manuscript provides a comprehensive overview of the current state of biogeochemical modeling in the context of operational oceanography. The authors have successfully outlined the complexity of these models, the various approaches used to couple biogeochemical processes with physical oceanographic models, and the challenges associated with parameterization and computational costs. The manuscript is well-structured, and the use of specific examples from different modeling systems enhances its clarity and relevance. The paper is a valuable contribution to the field, offering insights into the advancements and ongoing challenges in marine biogeochemical modeling.**
We thank the reviewer for her/his positive general comment

**Specific Comments**
**1. Introduction:**
**The introduction effectively sets the context for the discussion, presenting the main aspects of modelling. However, it would benefit from highlighting the importance of biogeochemical modelling for understanding marine ecosystems, giving more details on the practical applications and implications of these models for marine resource management and policymaking. The reader would like to know why modelling ocean biogeochemistry is important. What is at stake?**
We thank for the comment, we propose to add a sentence at old line 28:
By providing timely information for the current state and a consistent reconstruction of the past, biogeochemical operational models can support the estimation of ocean carbon sequestration and storage, monitoring effects of acidification and deoxygenation, marine spatial planning, and, as input for habitat and food web modelling, marine biodiversity conservation and fisheries management.

**2. Basic Formulation and Processes of Biogeochemical Models:**
**The section provides a clear and concise explanation of the fundamental principles behind biogeochemical models. Including a schematic diagram of a typical biogeochemical model could further enhance the readers' understanding.**
We will add a figure reporting a simplified scheme of biogeochemical models (Figure 1), and we will add a sentence at old line 67:
Figure 1 shows a schematic representation of the cycling of several chemical elements (e.g. nutrients) between living and non-living compartments, as well as some additional features typically included in biogeochemical models, which are presented below.

[Figure]

Fig. 1: simplified scheme of compartments (boxes) and processes (arrows) representing the cycles of multiple chemical elements typically included in a biogeochemical model. The symbol ... indicates increase of number of compartments (e.g., multiple plankton functional types, multiple size compartments of non-living organic matter or additional variables resolved by the carbonate system.

**The discussion on the NPZD approach is thorough, but it would be useful to compare this approach with other modeling frameworks, highlighting their respective strengths and limitations.**
We thank the reviewer for the comment, however, given the nature of the contribution (i.e., short book chapter on the status of operational biogeochemical models) and the fact that other chapters of this collection are presenting and discussing other model frameworks (e.g. ecosystem and high trophic level models in Libralato, 2024, we would avoid to include this part here.

Libralato, S. (2024). Numerical Models for Monitoring and Forecasting Ocean Ecosystems: a short description of present status. State Planet Discuss. [pre-print], https://doi.org/10.5194/sp-2024-42

**The manuscript explains the trade-offs between model complexity and computational cost. It would be beneficial to include a table summarizing the key features and applications of the various models discussed (e.g., HadOCC, SCOBI, NEMURO, etc.).**
We thank the reviewer for the comment. Models and operational systems are constantly evolving, so a better option than a static table, which would inevitably be incomplete and soon outdated, is to introduce the Atlas of Operational Systems collected as part of the OceanPredition Decade Collaborative Center (OceanPredition DCC, oceanpredition.ogs) initiative (Alvarez Fanjul et al., 2024). The OceanPredition initiative, of which this chapter is a part, aims to create a collaborative framework for defining common architectures and new tools, standards and best practices for operational ocean systems. The initiative proposes an atlas listing the operational systems where the description of models is kept updated.

We will change the paragraph at old lines 124-125 as follows:
As part of the UN Decade of Ocean Science for Sustainable Development program, the Ocean Prediction Decade Collaborative Center (Alvarez Fanjul et al., 2024) is promoting the Atlas of Operational Systems, which also describes their biogeochemical component (oceanprediction.org). Some of the biogeochemical models used in operational prediction systems are briefly presented below, roughly ordered by increasing complexity.

Alvarez Fanjul E., Ciliberti S., Pearlman J., Wilmer-Becker K., Bahurel P., Ardhuin F., Arnaud A., Azizzadenesheli K., Aznar R., Bell M., Bertino L., Behera S., Brassington G., Calewaert J. B., Capet A., Chassignet E., Ciavatta S., Cirano M., Clementi E., Cornacchia L., Cossarini G., Coro G., Corney S.,

Davidson F., Drevillon M., Drillet Y., Dussurget R., El Serafy G., Fearon G., Fennel K., Ford D., Le Galloudec O., Huang X., Lellouche J. M., Heimbach P., Hernandez F., Hogan P., Hoteit I., Joseph S., Josey S., Le Traon P. -Y., Libralato S., Mancini M., Martin M., Matte P., McConnell T., Melet A., Miyazawa Y., Moore A. M., Novellino A., O'Donncha F., Porter A., Qiao F., Regan H., Robert-Jones J., Sanikommu S., Schiller A., Siddorn J., Sotillo M. G., Staneva J., Thomas-Courcoux C., Thupaki P., Tonani M., Garcia Valdecasas J. M., Veitch J., von Schuckmann K., Wan L., Wilkin J., Zhong A., Zufic R.: Promoting best practices in ocean forecasting through an Operational Readiness Level. Frontiers in Marine Science, 11, 1443284, 2024

**The section on the incorporation of additional components (e.g., metals, POPs) into biogeochemical models is particularly interesting. More examples of how these components have been successfully integrated and validated in existing models would strengthen this section.**

This paragraph aimed to introduce additional elements that increase the complexity and realism of operational biogeochemical models with potential implications for societal applications. However, the maturity of models incorporating metals and POPs is not yet at a stage where they can be included in operational prediction systems, and they would merit a separate discussion which is beyond the scope of the present chapter.

We will clarify this point, also considering Rev#2 suggestion, by changing old lines 119-121 as follows:

Although not yet ready to be integrated into an operational prediction system, additional - and useful to society - components of biogeochemical models are the fate, dynamics and transport of metals and persistent organic pollutants (POPs; Melaku Canu et al., 2015; Wagner et al., 2019, Bieser et al., 2023), including bioaccumulation in low trophic level compartments and cumulative impacts on marine species and ecosystems (Rosati et al., 2022; Zhang et al., 2020).

**Even if it's not the focus of the paper, the discussion on data assimilation could be developed further. The manuscript could delve deeper into the specific challenges and potential solutions related to assimilating different types of observational data (e.g., satellite data vs. in situ measurements).**

We will add some additional details on assimilation of satellite data and in situ data, but without going too much into this topic and its challenges, since it deserves a separate discussion which would go beyond the scope of this chapter. Indeed, data assimilation is the subject of a separate chapter in this collection (Martin et al., 2024).

Operational biogeochemical models include also data assimilation schemes (Brasseur et al., 2009; Fennel et al., 2019), with satellite data being the most commonly used due to their spatial coverage and time availability, even in near real time. Ocean colour chlorophyll is the variable most typically assimilated in biogeochemical models (Nerger and Gregg, 2008; Ciavatta et al., 2011; Fiechter et al., 2011; Fontana et al., 2013; Teruzzi et al., 2014; Ciavatta et al., 2016; Song et al., 2016; Mattern et al., 2017) but other remote sensing variables have also been tested: diffuse attenuation coefficient (Ciavatta et al., 2014), phytoplankton functional type chlorophyll (Ciavatta et al., 2018; Skákala et al., 2018, Pradhan et al. 2020) and inherent optical properties (Jones et al., 2016). While ocean colour provides unique information about the surface of the ocean, the transfer of surface information to deeper layers usually requires approximations.

Then, the following comment's reply complements the aspects of the assimilation of insitu observations

Additional reference:

Martin, M., Hoteit, I., Bertino, L., Moore, A.M. (2024). Data assimilation schemes for ocean forecasting: state of the art. State Planet Discuss. [pre-print],https://doi.org/10.5194/sp-2024-20

Ciavatta, S., Torres, R., Martinez-Vicente, V., Smyth, T., Dall'Olmo, G., Polimene, L., & Allen, J. I.: Assimilation of remotely-sensed optical properties to improve marine biogeochemistry modelling. Progress in Oceanography, 127, 74-95, 2014

Ciavatta, S., Brewin, R. J. W., Skakala, J., Polimene, L., de Mora, L., Artioli, Y., & Allen, J. I.: Assimilation of ocean-color plankton functional types to improve marine ecosystem simulations. Journal of Geophysical Research: Oceans, 123(2), 834-854, 2018

Skákala, J., Ford, D., Brewin, R. J., McEwan, R., Kay, S., Taylor, B., De Mora, L. & Ciavatta, S.: The assimilation of phytoplankton functional types for operational forecasting in the northwest European shelf. Journal of Geophysical Research: Oceans, 123(8), 5230-5247, 2018

Pradhan, H. K., Völker, C., Losa, S. N., Bracher, A., & Nerger, L.: Global assimilation of ocean-color data of phytoplankton functional types: Impact of different data sets. Journal of Geophysical Research: Oceans, 125(2), e2019JC015586, 2020

Jones, E. M., Baird, M. E., Mongin, M., Parslow, J., Skerratt, J., Lovell, J., Margvelashvili, N., Matear, R. J., Wild-Allen, K., Robson, B., Rizwi, F., Oke, P., King, E., Schroeder, T., Steven, A., and Taylor, J.: Use of remote-sensing reflectance to constrain a data assimilating marine biogeochemical model of the Great Barrier Reef, Biogeosciences, 13, 6441–6469, https://doi.org/10.5194/bg-13-6441-2016, 2016.

**The role of emerging technologies, such as autonomous underwater vehicles and biogeochemical Argo floats, is briefly mentioned. Expanding this discussion to include recent advancements and case studies would provide a more comprehensive view.**

Thanks, focusing on operational biogeochemical systems and avoiding a too much expansion of this part, we will include details on the in situ variables that are currently assimilated in, and used for validation and calibration of operational biogeochemical model systems.

We will change old lines 53-55 as follows:

The emergence of autonomous underwater sensors (biogeochemical gliders and Argo floats) has opened the possibility to better constrain biogeochemical dynamics also below the water surface (Verdy and Mazloff, 2017; Cossarini et al., 2019; Teruzzi et al., 2021; Skákala et al., 2021). Oxygen, chlorophyll, nitrate profiles are currently used in forecast system for assimilation (Amadio et al., 2024), but also used for parameters tuning (Wang et al., 2021, Yumruktepe et al., 2023, Falls et al., 2022), validation of operational systems (Salon et al., 2019; Mignot et al., 2023) and adaptive monitoring of phytoplankton blooms (Ford et al., 2022).

Additional reference to be included:

Mignot, A., Claustre, H., Cossarini, G., D'Ortenzio, F., Gutknecht, E., Lamouroux, J., Lazzari, P., Perruche, C., Salon, S., Sauzède, R., Taillandier, V., and Teruzzi, A. (2023). Using machine learning and Biogeochemical-Argo (BGC-Argo) floats to assess biogeochemical models and optimize observing system design. *Biogeosciences*, *20*(7), 1405-1422.

Wang, B., Fennel, K., Yu, L., and Gordon, C. (2020). Assessing the value of biogeochemical Argo profiles versus ocean color observations for biogeochemical model optimization in the Gulf of Mexico. *Biogeosciences*, *17*(15), 4059-4074.

Yumruktepe, V. Ç., Mousing, E. A., Tjiputra, J., and Samuelsen, A. (2023). An along-track Biogeochemical Argo modelling framework: a case study of model improvements for the Nordic seas. *Geoscientific Model Development*, *16*(22), 6875-6897.

Falls, M., Bernardello, R., Castrillo, M., Acosta, M., Llort, J., and Galí, M. (2022). Use of genetic algorithms for ocean model parameter optimisation: a case study using PISCES-v2_RC for North Atlantic particulate organic carbon, Geosci. Model Dev., 15, 5713–5737, https://doi.org/10.5194/gmd-15-5713-2022, 2022.

Ford, D. A., Grossberg, S., Rinaldi, G., Menon, P. P., Palmer, M. R., Skakala, J., Smyth, T., Williams, C.A.J., Lopez, A.L. and Ciavatta, S. (2022). A solution for autonomous, adaptive monitoring of coastal ocean ecosystems: Integrating ocean robots and operational forecasts. Frontiers in Marine Science, 9, 1067174.

Verdy, A., and Mazloff, M. R. (2017). A data assimilating model for estimating Southern Ocean biogeochemistry. Journal of Geophysical Research: Oceans, 122(9), 6968-6988.

**The title of this section is a little reductive in relation to all the aspects covered. I suggest rewording the title or dividing it into several subsections (ex: basic formulation, coupling, model complexity, data assimilation, main models used...).**
Thanks for the suggestion, we will propose to subdivide the section in two subsections with the following titles:
2.1 Formulations, processes and elements of biogeochemical models
2.2 Main models used in operational systems

**3. Conclusions:**
**The conclusions summarize the key points effectively but could be expanded to include a forward-looking perspective on the future of biogeochemical modeling. What are the emerging trends and technologies that could shape this field in the next decade?**
**Discussing the potential for interdisciplinary collaboration and the integration of biogeochemical models with socio-economic models could provide a broader context for the readers.**
We will complement the last part of the conclusion by adding the following points:
Rapidly evolving applications of artificial intelligence in marine biogeochemistry can assist in optimising model parameters, developing hybrid models to improve predictions and operational system efficiency, and detecting patterns in large data sets from reanalysis. Linking microbial community dynamics to ecosystem processes through metagenomic data can improve models describing nutrient cycling, carbon fluxes and diversity. New coupling paradigms are needed to promote the integration of biogeochemical models with the dynamics of pollutants, high trophic levels and Earth system components. In addition to science-driven developments, operational biogeochemical systems can evolve to respond to societal demands and user requests reflecting the needs for robust, accurate and actionable insights about the functioning, health and biodiversity of marine ecosystems, the link to sustainable fisheries and aquaculture, the assessment of the compound impacts of heat waves, oxygen depletion and acidification on marine resources and the role of the oceans in achieving the goal of carbon neutrality.

**Technical Corrections**
**Introduction:**
**- The definition of 'reanalysis', 'hindcast' and 'nowcast' may vary depending on the application, methodology, community or context. A definition of these terms should be provided.**
We would like to propose the following definitions as described in Fennel et al., 2023:
"**Hindcasts** are model applications where a defined time period in the past was simulated. They can be unconstrained – in the sense that no observations are fed into the model except for initial, boundary, and forcing conditions – or constrained, where observations inform the model state via data assimilation. The latter are also referred to as optimal hindcasts or **reanalyses**. **Nowcasts/forecasts** are similar to constrained hindcasts but with the simulations carried out up to the present (referred to as nowcasts) or into the future (referred to as forecasts). The latter require assumptions about future forcing and boundary conditions, e.g., from other forecasts or climatologies or assuming persistence."

Therefore, the sentence at old lines 25-28 will be changed as follows:
Operational biogeochemical models have generally been developed by incorporating process-based biogeochemical models developed for research into existing physical forecasting systems (Gehlen et al., 2015) and are used to assess (i) past and current marine ecosystem states and trends (reanalysis, hindcast and nowcast) and (ii) short-term (days to weeks) or seasonal (months) forecasts (Le Traon et al., 2019; Fennel et al., 2019). When observations are assimilated, simulations of a past period are called reanalysis, while unconstrained simulations of a past period are called hindcast. When the simulations are carried forward to the present, they are referred to as nowcasts. (Fennel et al., 2023)

Fennel, K., Long, M. C., Algar, C., Carter, B., Keller, D., Laurent, A., Mattern, J. P., Musgrave, R., Oschlies, A., Ostiguy, J., Palter, J. B., and Whitt, D. B.: Modelling considerations for research on ocean alkalinity enhancement (OAE), in: Guide to Best Practices in Ocean Alkalinity Enhancement Research, edited by: Oschlies, A., Stevenson, A., Bach, L. T., Fennel, K., Rickaby, R. E. M., Satterfield, T., Webb, R., and Gattuso, J.-P., Copernicus Publications, State Planet, 2-oae2023, 9, https://doi.org/10.5194/sp-2-oae2023-9-2023, 2023

**- Line 30, cite some pollutants in brackets to help the reader.**
The sentence at old line 30 will be changed as follows:
They can also include carbonate systems, pollutants (e.g. Hg, persistent organic pollutants -POPs) and other features…

**Equation Formatting: Ensure that all equation terms are consistently formatted and clearly presented. For example, line 40, KH should be written KH (with H as a subscript). Please check all terms (KV, Rbio...).**
Thanks for noting. $K_H$, $K_V$ and $R_{BIO}$ will be correctly presented in the text.

**References: Please check that all works cited are included in the reference list and that all references are correctly cited in the text. Please also check the spelling of the names mentioned and the year of publication. There are many typos... This list below is not exhaustive, please check each reference...**
**-The text refers to Daewl instead of Daewel, Gutkenecht instead of Gutknecht.**
**-Vichi et al (2007), Vichi (2015), Yumruktepe et al (2023) are cited in the text but not included in the reference list.**

**-Vichi et al (2017), Yumruktepe et al (2022) are mentioned in the Reference list but are not cited in the text.**
We will check carefully the references

**Acronyms: All acronyms and chemical products should be defined when first used. For example: DMS HTL, POPs, CDOM...**
Thanks, all acronyms will be defined when first used.
Dimethyl sulfide (DMS), Persistent organic pollutants (POPs), higher trophic level (HTL), coloured dissolved organic matter (CDOM)

**Typographical Errors**
**Line 26: " process-base" should be " process-based".**
Done

**Line 38: "compartmens" should be "compartments"**
Done

**line 90: "plankton functional groups" should be "plankton functional types" to be consistent with the acronym PFTs**
Done

**Line 141, remove the dot before the reference Aumont et al (2015)**
Done

**line 150, please check: "multinutrients and multi-plankton several", something is wrong**
Corrected as follows:
with multinutrients, eight plankton functional types and …

---

## Author Comment (AC2)

**Title: Numerical Models for Monitoring and Forecasting Ocean Biogeochemistry: a short description of present status**
Author(s): Gianpiero Cossarini, Andy Moore, Stefano Ciavatta, Katja Fennel
MS No.: sp-2024-8
Report: Ocean prediction: present status and state of the art

Dear Editor,
We thank the 3 reviewers for their positive comments. Their requests have been carefully considered. In the following reply letter, reviewers' comments are in **bold green**, the answers in black and the proposed revised text in blue.

**RC2: 'Comment on sp-2024-8', Anonymous Referee #2, 22 Oct 2024 reply**
**The authors present a review of the state-of-the-art in biogeochemical modelling, covering various aspects such as usage, complexity, limitations, cost. A list of common biogeochemical models is also provided.**
**All in all, the paper covers the topics that one would expect to find when first learning about BGC models, and thus fulfills its role as a chapter in the special issue (if I understood correctly).**
**I have only a few minor comments:**
**\* line 45, some modelled processes may not conserve mass, if some of the implied variables are not one of the model state variables (e.g. denitrification in the Bamhbi model). Thus, the sentence at it is now may be misleading**
Conservation of mass is a solid principle for tracers included in the $R_{bio}$ term of a biogeochemical model. We will change the sentence as follows:
... and is typically based on the principle of conservation of mass ...

**\* line 78: define DMS, unless it is defined in previous chapters of the special issue ?**
The acronym will be defined when first used: dimethylsulfide (DMS)

**\* line 102: "most fitted" makes me think of fitting a curve through some points (in a statistical sense). Maybe you meant "most fit" or "fittest" (species) ?**
The sentence will be revised as follows:
 ... so that the fittest groups can prevail in the resulting ecosystem.

**\* line 119: define POP**
The acronym will be defined when first used: persistent organic pollutants -POPs

**\* line 150: remove "several"**
Done

**\* model list: consider adding Darwin to the list (the BGC model associated with MiTgcm) ?**
We agree Darwin is worth to be listed and described. We will add the followings sentence:
DARWIN: a complex multi-nutrient model in which the plankton community comprises hundreds of groups by taking into account cell size and functional traits to study biodiversity and biogeography (Dutkiewicz et al., 2009).

Dutkiewicz, S., Follows, M. J., & Bragg, J. G.: Modeling the coupling of ocean ecology and biogeochemistry. Global Biogeochemical Cycles, 23(4), 2009

---

## Author Comment (AC4)

**Title: Numerical Models for Monitoring and Forecasting Ocean Biogeochemistry: a short description of present status**
Author(s): Gianpiero Cossarini, Andy Moore, Stefano Ciavatta, Katja Fennel
MS No.: sp-2024-8
Report: Ocean prediction: present status and state of the art

Dear Editor,
We thank the 3 reviewers for their positive comments. Their requests have been carefully considered. In the following reply letter, reviewers' comments are in **bold green**, the answers in black and the proposed revised text in blue.

**RC3: 'Comment on sp-2024-8', Anonymous Referee #3, 25 Oct 2024 reply**
**Review of "Numerical Models for Monitoring and Forecasting Ocean Biogeochemistry: a short description of present status"**
**By Gianpiero Cossarini, Andy Moore, Stefano Ciavatta, and Katja Fennel**
**The paper provides an overview of the biogeochemical models used for operational oceanography today. It provides basic information about how marine biogeochemical models are coupled with ocean general circulation models and discusses uncertainties related to parameterisations, initial conditions, and the lack of observations. The paper is well written and organized. With the understanding that is is not a classical science paper, but an overview in the context of a report on operational oceanography, I have only a few comments and corrections listed below:**
We thank the reviewer for her/his positive comment

**Specific comments**
**Line 32: "… while detailed descriptions and discussions can be found in the following articles ". It sounds like these three papers are the definitive list to read if you want to read all about models in operational oceanography, I suggest to change "while" with "more".**
Done

**Line 57: "Rather, equations describing biogeochemical processes rely on empirical relationships based on laboratory experiments (e.g., nutrient limitation experiments, grazing dilution experiments), biological theories, and ecological principles based also on biogeographic relationships." Here you should also mention conservation of matter, which is one solid principle that can be applied in these models.**
The principle of conservation of mass is introduced at old lines 44-45 as follows:
The last term, $R_{bio}$, represents the local source-minus-sink terms for the biogeochemical tracers and is typically based on the principle of conservation of mass to simulate the cycling of chemical elements through various marine compartments.

**Line 97: I am uncertain about what you mean by this regarding the subdivision of zooplankton "and its role within an end-to-end ecosystem approach (Mitra et al., 2014)." Is this with respect to who eats them?**
We agree, the sentence was not clear and it will be removed. Mixotrophy is introduced in the next sentence.

**Line 119: I suggest to include this paper by Bieser et al., 2023 in the reference list (https://doi.org/10.5194/gmd-16-2649-2023)**
Thanks for the suggestion. The new reference will be added.

Bieser, J., Amptmeijer, D. J., Daewel, U., Kuss, J., Soerensen, A. L., and Schrum, C.: The 3D biogeochemical marine mercury cycling model MERCY v2.0 – linking atmospheric Hg to methylmercury in fish, Geosci. Model Dev., 16, 2649–2688, https://doi.org/10.5194/gmd-16-2649-2023, 2023.

**Line 122: I think there are quite a few physical models (even if there are more biogeochemical models). I think it would be better to highlight that physical models solve the same equations, but differ mainly in how they are discretized on the horizontal and vertical grid. Physical models also differ in how they parameterized subgridscale processes. Biogeochemical models, on the other hand, solve entirely different sets of equations, in addition to being discretized on different grids and having to parameterize processes that are not included explicitly.**
The aim here is only to present the large variety of biogeochemical models used in operational systems, while the number of dynamic ocean models is relatively smaller. A complete list of physical models (and a discussion on their characteristics) is the goal of another chapter of this collection (Bell et al., 2024). We would prefer not to change this sentence unless the reviewer suggests it.

Bell, M.J., Drillet, Y., Martin, M., Schiller, A., Ciliberti, S. (2024). Numerical Models for Simulating Ocean Physics. State Planet Discuss. [pre-print], https://doi.org/10.5194/sp-2024-41

**Line 140: PICES also exists with variable stoichiometry (PISCES-QUOTA), but the version used operationally uses constant stoichiometry, this should probably be mentioned.**
Thanks for the suggestion. We will add a sentence on the PISCES-QUOTA version:
A version with variable stoichiometry (PISCES-QUOTA) also exists and is used for climate scenario studies (Kwiatkowski et al., 2018).

Kwiatkowski, L., Aumont, O., Bopp, L., & Ciais, P.: The impact of variable phytoplankton stoichiometry on projections of primary production, food quality, and carbon uptake in the global ocean. *Global Biogeochemical Cycles*, *32*(4), 516-528., 2018

**Technical corrections/language**
**Line 40-41: Use subscript H and V in KH and KV.**
Done

**Line 49: Suggest: "Different schemes can be used to couple the physical and biogeochemical processes to optimize accuracy and computational cost (Bruggeman and Bolding, 2014; Cossarini et al., 2017)."**
Done

**Line 78: to define = for**
Done

**Line 108: The microbial…**
Done

**Line 117: Just nekton ("organism" is unnecessary)**
Done